# Demographic, Knowledge and Impact Analysis of 57,627 Antibiotic Guardians Who Have Pledged to Contribute to Tackling Antimicrobial Resistance

**DOI:** 10.3390/antibiotics8010021

**Published:** 2019-03-09

**Authors:** Sophie Newitt, Olaolu Oloyede, Richard Puleston, Susan Hopkins, Diane Ashiru-Oredope

**Affiliations:** 1Field Service, Public Health England, Nottingham NG2 4LA, UK; sophie.newitt@phe.gov.uk (S.N.); richard.puleston@phe.gov.uk (R.P.); 2HCAI & AMR Division, Public Health England, London NW9 5EQ, UK; Olaolu.oloyede@phe.gov.uk (O.O.); susan.hopkins@phe.gov.uk (S.H.)

**Keywords:** antibiotic resistance, public health, campaign evaluation, antimicrobial stewardship, Antibiotic Guardian, behaviour change

## Abstract

In 2014, Public Health England (PHE) developed the behavioural change Antibiotic Guardian (AG) campaign to tackle antimicrobial resistance (AMR). This included an online pledge system aimed at healthcare professionals (HCP) and the public. Demographics of AGs were collected when pledging online and analysed by pledge group, type, geography, and source of hearing of the campaign between 24/07/2014–31/12/2017. Website visitors and acquisition routes were described using Google analytics data. From November 2016, five questions assessed AMR knowledge which was compared to published Eurobarometer AMR survey results for UK. Behaviour change of AGs was also assessed through an impact questionnaire, evaluating the effect of the campaign on self-reported behaviour around AMR. Overall there were 231,460 unique website visitors from 202 countries resulting in 57,627 English and 652 foreign language pledges. Website visitors increased each year with peaks during European Antibiotic Awareness Day and (EAAD) World Antibiotic Awareness Week (WAAW). Self-direction was the largest acquisition route (55%) with pledges more likely via this route than social media (OR 2.6, 95% CI 2.5–2.6). AGs (including the public) were more likely to answer questions correctly than the Eurobarometer UK group (OR 8.5, 95% CI 7.4–9.9). AG campaign engagement has increased over the four years with particular increases in the student group. AGs had greater knowledge compared to the Eurobarometer UK population. The latest impact evaluation of the online pledge scheme highlights that it continues to be an effective and inexpensive way to engage people with the problem of AMR especially among those with prior awareness of the topic.

## 1. Introduction

Antibiotic therapy has been one of the most important medical discoveries of the 20th century, preventing millions of deaths worldwide from infections and facilitating advancement in modern medicine [1]. Antimicrobial resistance (AMR) is a growing concern and now recognised as one of the significant public and global health threats of our time. The Antibiotic Resistance Threat Report (2013) in the United States estimated at least 2 million people each year acquire antibiotic resistant infection, with about 23,000 dying as a result [2]. Antibiotic resistance challenge requires a coordinated approach with countries and across sectors such as the environment, human and animal health [3]. The World Health Organization (WHO) in 2015 published the Global AMR Action Plan (GAP) which outlined five strategic objectives [4]. Improving awareness and understanding of AMR is one of the five strategic objectives of this plan [4] as well as several national action plans on antimicrobial resistance including the UK AMR Strategy [5]. This is further supported by the published AMR review which called for a worldwide awareness campaign following an estimation in 2014 that there were 700,000 deaths worldwide attributed AMR [6].

In 2014, Public Health England (PHE) developed the behaviour change and engagement campaign, Antibiotic Guardian (AG) in the UK with the aim of tackling AMR [7]. This included a dedicated website (www.antibioticguardian.com) with an online pledge system designed to improve knowledge and behaviours regarding antibiotic prescribing and use among healthcare professionals and the public.

As part of the inexpensive method of developing and implementing the campaign, additional promotional methods were used. This included:A 2-min video which was aimed at the public, embedded at the top of the campaign website and hosted on YouTube.Active promotion of key messages through Social media in particular Twitter, actively promoting the use of hashtag #AntibioticGuardian.Development of a range of resources (leaflets/posters) and toolkits available for healthcare professionals to use in local campaigns/outreach (signposted to through the website https://antibioticguardian.com/healthcare-professionals/).Development of an interactive quiz on the online publishing platform Playbuzz (http://antibioticguardian.com/Resources/playbuzz/).

A process evaluation after six months found that the campaign had exceeded its initial goal to engage and obtain 10,000 Antibiotic Guardian pledges from healthcare professionals and members of the public by 30 November 2014; less than six months after the campaign started [8,9]. A further impact evaluation carried out after the first year highlighted that the campaign increased knowledge and behaviour change (self-reported), as well as increased commitment to tackling AMR [10].

To ensure the continued success of the Antibiotic Guardian campaign, we examined the campaign after four seasons (years) to assess whether engagement had continued, describe the demographics and knowledge of AGs and to highlight any areas that could be addressed in future promotional activities.

## 2. Results

### 2.1. Engagement

Since the start of the campaign, the website has been visited 291,431 times of which 79% were unique visitors (231,460). These visits have translated into 57,627 English language pledges, with an unadjusted conversion rate of 19.8% for the website as a whole. The number of visits to the site and pledges has increased each year (Table 1). However, the conversion rate from 2014 has been relatively stable with the exception of a decline in 2016 to 15.6%. A similar decrease was also observed when the conversion rate was adjusted only to include new users. Translated website pages were available from November 2016 with 627 foreign language pledges and received the following conversion rates; Russian webpage 8.5%, Dutch webpage 19.3% and French webpage 8%.

The website received unique visits from 202 countries. The majority of these visits were from the UK (80.9%), but 19.1% of visits were from outside of the UK. The top five countries visiting the website outside of the UK were United States (n = 7391), Russia (n = 4675), Belgium (n = 3071), India (n = 1496) and Australia (n = 1420).

Overall, 44.6% of pledges were made from a unique IP address whereas 55.4% were from a repeated IP address. The median number of pledges made from a repeated IP address was 192 and ranged from 2–1647. The number of pledges from repeat IP addresses showed an increase from 64.2% in 2015 to 66.4% in 2017 (chi = 1200, *p* < 0.001). 8.7% (5019) of AGs had pledged more than once between 2014 and 2017.

For each year, there was a peak in the number of English pledges received during WAAW/EAAD which occur yearly in November (Figure 1). A small peak in pledges was also noted in March 2016. However, the number of pledges during WAAW in 2017 (4682), decreased compared to 2016 (5120). The number of organisations in the UK who registered planned activities to be carried out during WAAW to raise awareness of antibiotic resistance in 2014 was 187, with a peak of 305 organisations in 2015 and 103 in 2016. In 2017, 149 organisations registered planned activities; leading to a 44.7% increase in registrations over the prior year.

Up to the end of 2017, where country was stated (n = 43,065) pledges had been received from 149 different countries therefore antibiotic guardians were present in 57.7% of countries worldwide; 95.9% (n = 41,309) from the UK, 1.4% (n = 618) from the rest of Europe and 2.6% (n = 1138) from the rest of the world. There was an increase in the number of countries with AGs from 63 countries in 2015 to 149 (Country map of AG—Figure 2) in 2017 (137% increase). Country information was not collected during 2014.

The rate of AGs per 100,000 population in the UK between 2014–2017 was 83.3 (95% CI 82.6–84). There was some variation between UK countries with Wales having the highest rate at 206.6 per 100,000 population (95% CI 201.6–211.7), followed by England 81.1 (95% CI 80.3–81.9), Scotland 51.6 (95% CI 49.7–53.6) and Northern Ireland 35.2 (95% CI 32.6–38). A further breakdown by clinical commissioning groups (CCGs) and Health boards can be seen in Figure 3 and Appendix A. Rates by local government authorities can be found in the Appendix A which ranged from a rate of 9.1 (95% CI 5.2–14.8) per 100,000 population to 457.9 (95% CI 427.9–489.5) per 100,000 population.

The majority of unique visitors accessed the site through a desktop computer (58.8%) compared to a mobile device (e.g., phone or tablet) (41.2%). Individuals were also more likely to pledge to become an Antibiotic Guardian if accessing the site via a desktop computer compared to a mobile device, OR 1.26 (95% CI 1.23–1.28).

The most common acquisition route for unique visits to the site was self-directed, e.g., through a search engine or typing the website address directly into the browser which accounted for 55.2% of visits compared to 24.6% referred through social media, 15.3% referred through another website and 1.2% referred through links in emails. Other routes accounted for 3.7% of unique visits. Self-direction has remained the largest acquisition route across all four years. Compared to referrals via social media, individuals were more likely to pledge is they accessed the site through self-directed (OR 2.56, 95% CI 2.49–2.63) or via email referrals (OR 2.5, 95% CI 2.28–2.73) but showed no real difference of likelihood to pledge when referred through other websites (OR 0.99, 95% CI 0.96–1.04).

From 2014 to July 2016 there had been 136 publications, 108 bulletins/newsletters, 294 web articles and 40 international mentions of Antibiotic Guardian. The educational YouTube video entitled ‘Will you be an Antibiotic Guardian?’ accessible via the AG website and PHE YouTube channel was viewed 35,827 times with an average view time of 1.33 min out of 2.02 min (76% of the full video) and the ‘Antibiotic Resistance Playbuzz’ quiz launched on the website in November 2016 has been accessed 3900 times with 628 completions up until December 2016. Analysis of Twitter data over the first four years showed that #AntibioticGuardian had been tweeted 46,152 times by 15,307 people and 10,101 times during WAAW 2016. In 2017 WAAW 2333 individuals participated in Twitter’s social media with 5737 tweets posting the #AntibioticGuardian hashtag.

### 2.2. Audience

The most common pledge group overall was health care professionals (54.3%) (Table 2), which has remained the highest through the four years. Within this group, the most common pledge groups are pharmacy teams (34%), nurses (20.8%) and primary/secondary prescribers (16.8%) (Figure 4). The proportion of AGs who were nurses increased from 7.4% (n = 908) in 2014 to 13.4% (n = 2030) in 2017. Dentists also increased from 1.7% (n = 215) to 2.8% (n = 428). The most common pledge chosen by health professionals was ‘I will check that antibiotic prescriptions comply with the local guidance and query those that do not’. This accounted for 13.4% of pledges (n = 4155) and was one of the pharmacy pledges. A further breakdown of the most common pledge for each sub group of the three main pledge groups is shown in Table 3.

There was a significant increase in the proportion of AG that are in student and educator pledge category from 8.8% in 2014 to 19.4% in 2017 (chi = 947.13, *p* < 0.001). The highest proportion of students pledging were pharmacy students (Table 4. The most common pledge choice for students was ‘the next time I see an antibiotic prescribed, I will ask the prescriber about the indication and duration, to understand if this is in accordance to local and national guidelines’ which accounted for 19.2% of pledges (n = 1767). This was the most common pledge chosen by healthcare students and non-healthcare students.

There was a reduction in the proportion of pledgers who are members of the public (28% in 2014 vs. 23.8% in 2017) which was significant (chi = 277.4, *p* < 0.001), the majority of which were adults (79.7%). The most common pledge chosen by the public was ‘for infections that our bodies are good at fighting off on their own, like coughs, colds, sore throats and flu, I pledge to try treating the symptoms for five days rather than going to the GP’. This was an adult sub group pledge and accounted for 32.3% of pledges (n = 5211). Where gender could be inferred (n = 51,149), the majority of AGs were female (73%).

### 2.3. AMR Knowledge

In total 3289 AGs answered 16,445 questions on AMR knowledge (5 questions each); 94.4% (n = 15,521) were answered correctly and 80% of AGs answered all five questions correctly. For individual questions, the percentage answered correctly ranged from 91.1% for ‘taking antibiotics often has side effects such as diarrhoea’ to 98% for ‘unnecessary use of antibiotics makes them become ineffective’ (Table 5).

There was some variation between pledge groups for the percentage of questions answered correctly as shown in Table 5. The HCP answered significantly more questions correctly (96.5%, n = 8677/8995) compared to students (92.8%, n = 3375/3635) (chi 154.64, *p* < 0.001) and the public (91.2%, n = 3469/3805) (chi = 77.57, *p* < 0.001).

The Eurobarometer survey on Antimicrobial resistance used questions 1 to 4 to survey the EU general population and publish results on AMR knowledge by country. For each of these questions, all AG groups (including the public) answered a higher proportion of these correctly when compared to the UK and EU published figures (Table 5). AGs were more likely to answer all four questions correctly than the UK group (OR 8.5, 95% CI 7.4–9.9) and the EU group (OR 13.9, 95% CI 12.7–15.3). AGs answered an average of 3.8 questions correctly compared to 2.8 for UK and 2.5 for EU.

### 2.4. Source

Where indicated (n = 46,798), AGs (HCPs, students and public combined) most commonly heard about the campaign from colleagues (23.9%), within the NHS (23.3%) or from professional organisations (14.8%) (Figure 5). These sources remained the most common sources all through 2014 to 2017. However, increases were noted in hearing through the university (2015 and 2016) and social media (2016).

The source varied by group with the public mainly hearing about the campaign by NHS (31.1%) and the social media (20.9%), HCPs through colleagues (33.2%) and professional organisations (20.5%), whereas students heard from the university (47.6%). The public was more likely to hear about the campaign from social media than the health care professionals (OR 3.10, 95% CI 2.91–3.31) and students (OR 1.94, 95% CI 1.36–1.61).

### 2.5. Antibiotic Guardian Campaign—Assessment of Self Reported Knowledge and Behaviour Change

Overall, 1940 AGs (6.8% response rate) completed the online survey which self-reported behaviours following their pledges (outcomes). Of those with gender recorded, the majority (76.8%) were females and about half (56%) of the respondents pledged as health professionals with 30.8% (598) pledging as a member of the public. Of those respondents, where the year of the pledge was indicated (n = 1031), almost all (95.1%) had pledged within the previous 16 months of the survey. Characteristic of AGs that participated in the survey is shown in Table 6.

Prior to hearing about the AG campaign, 86% of respondents were aware of antibiotic resistance, 32.2% (537) of this group stated that they believed that antibiotic resistance is an important issue they can do something about as individuals. Where indicated (n = 1696), 28.1% believed Antibiotic Guardian website was an important source that impacted their knowledge on antibiotic resistance.

In selecting a pledge, 47.5% of respondents (922) chose a specific pledge because of their belief in successfully delivering on their commitment, of which 64.3% (593) were healthcare professionals. Of this group (n = 922), 60.6% self-reported to ‘always’ act in line with their pledge and 31.8% acted ‘most of the time.’ Overall, 61.8% of AGs ‘always’ acted in line with their pledge after the campaign. A higher proportion of AGs who were members of the public (84.9%), reported ‘always’ acting in line with their pledge compared to healthcare professionals (49.6%), with an odds ratio of 5.7 (95% CI 4.4–7.4, *p* < 0.001).

For members of the public, ‘I know that antibiotic resistance is a very important problem for public health’ (64.6%) was the most common reason for pledging to become an Antibiotic Guardian, while professional experience with antibiotic resistance (37.6%) was the most commonly stated reason for healthcare professionals.

## 3. Discussion

The Antibiotic Guardian behavioural change campaign aimed at improving AMR related knowledge and behaviour amongst healthcare professionals and the general public has been shown to have sustained engagement with increases in AG pledges over the years. As a low-cost campaign reliant on local and individual engagement and promotion activities, this pledge based approach shows the impact of alternatives to traditional advice based interventions for effective public health intervention programmes [11]. Findings gathered from the impact evaluation of the AG campaign were similar to previous findings [10], demonstrating that the campaign increased commitment to tackling AMR in both healthcare professional and member of the public. It also showed increased self-reported knowledge and changed behaviour particularly among people with prior AMR awareness. With members of the public more likely to act according to their pledge than healthcare professionals, the Antibiotic Guardian campaign as an online pledge scheme continues to be an effective and inexpensive way to engage people with the problem of AMR especially among those with prior awareness of the topic.

The number of pledges chosen yearly remains consistent with peaks in November during WAAW and EAAD due to the focused national, European and worldwide promotional activities. However, there has been a shift away from these profound peaks to an increase in year-round AG pledges and a decreasing trend in the number of pledges during WAAW compared to 2015.

Health care professionals continued to be the main groups engaged with the campaign. This may be due to promotion of the campaign historically focused on changing healthcare professional’s behaviours especially prescribing. However, since 2016 promotional activities have been focused on actively engaging students and the public by encouraging students from universities to nominate themselves or be nominated as local Antibiotic Guardian champions to run campaigns within their universities. This change has led to increases in the number of pledges from students and the public compared to 2014. There was also an increase in nurses and dentists coinciding with the publication (November 2016) of the dental antimicrobial stewardship toolkit [12]. It is encouraging that the campaign is engaging the general population which may therefore help ensure that addressing AMR is everyone’s responsibility rather than just the HCP. The increase in student pledges is also important as they will have the opportunity to influence the future healthcare workforce.

Engagement via social media was found to be particularly useful for the public with the majority of AGs from the public hearing about the campaign from this source. This highlights the need for continued social media promotion. A peak in pledges was also observed during March 2016 which corresponded with the use of paid for advertising used for the first time on Facebook to target mothers of young children and healthcare students aged 18–25 within England. This accentuated the potential of social media through dissemination of health campaign related information to the public in supporting behavioural change and engagement. Engagement through social media provides an opportunity to exchange on health matters and real-time learning [13].

The majority of website visitors and pledges were from the UK, but there was variation by country within the UK and on a local level which requires further understanding of different engagement and promotional activities used. Antibiotic Guardian rates per local authority have been published in response to the NICE guidelines Antimicrobial stewardship: changing risk-related behaviours in the general population aimed at local authorities [14]. Analysis of IP addresses found new pledges were regularly made from repeat addresses indicating one computer or organisation being used for multiple pledges. NHS organisations often have one IP address (although on multiple sites) so this fits with information from registered activities and social media of coordinated activities during WAAW with hospitals and universities increasing awareness of the campaign and making a computer or tablet available for people to sign up and pledge on the spot.

The campaign has also gathered interest across Europe with WHO-Europe and Belgian Antibiotic Policy Coordination Committee (BAPCOC) requesting collaboration with PHE to translate the current UK Antibiotic Guardian campaign into Russian, Dutch and French languages during 2016 [15].

Knowledge regarding AMR was found to be greater in AGs when compared to the Eurobarometer survey of the UK population and the EU. This also held true when focused only on the AGs who were members of the public. This is encouraging and could be an indicator of the knowledge acquired through the campaign due to the specific educational materials such as videos and quizzes available. However, it may also reflect a bias in the campaign towards those that already have an interest and are engaged in AMR. There were still some gaps in knowledge highlighted in all groups and in particular around knowing that antibiotics often have side-effects. Overall the general public had the least knowledge out of the AG groups, but this was still greater than the Eurobarometer comparison groups. This further illustrates the need for greater awareness regarding antibiotics and the continued efforts of such campaigns.

This study did not make any direct comparisons with prescribing data; this is in part due to the known multifactorial influences on prescribing and other efforts that are ongoing to reduce prescribing making causality challenging to conclude. However, it was found that the CCG in England that has made the most substantial reduction in antibiotic prescribing over the last two years (Birmingham CrossCity CCG) also had the highest number of AGs during 2016 (see Case Study—Figure 6).

Preliminary analysis also found a weak positive correlation between the rate of AG per 100,000 population (2014–2017) and the percentage reduction in the twelve-month rolling total number of prescribed antibiotic items per STAR-PU from December 2014 to December 2017 for Clinical Commissioning Groups within England (*p* = 0.06, R^2^ = 0.02).

The use of Google analytics data has provided greater insight into how the AG website has been used in order to target promotional activities. However, we are unable to describe those individuals that viewed the website but did not choose to pledge or have an understanding as to why they may not have pledged. It is unknown whether any AGs took part in the Eurobarometer studies and whether the study group was truly representative of the UK general public knowledge.

Further research will be needed to explore the difference between groups for a more targeted campaign strategy.

## 4. Methods

This was a retrospective cohort study to describe key measures of the Antibiotic Guardian campaign over four years, between 24/07/2014 to 31/12/2017. The number of overall visitors and unique visitors (new visitors) to the campaign website were analysed using Google analytics to compare numbers over time, geography, conversion rates, and acquisition routes [16]. The key data analysed are defined in Table 7 [8,16].

The primary outcome measure was pledging to become an Antibiotic Guardian in the English language on the website (www.antibioticguardian.com) (Figure 7). A full analysis of foreign language pledges (Dutch, French, Russian) was not included in this analysis as these were implemented as part of a pilot project in 2016 and described in a separate paper [15]. Demographic data was collected from Antibiotic Guardians when pledging online. This was used to calculate the number and proportion of AGs by pledge group, pledge type, geography, IP address, and source of hearing of the campaign in total and by the calendar year. Where possible, the sex of AGs was inferred by using the title provided when pledging, e.g., Mr, Ms, Miss. Exact matches of first name and surname were used to identify individuals who had pledged multiple times.

From November 2016, Antibiotics Guardians were asked to answer true or false to five statements on AMR to assess their knowledge using pop up website box after pledging to become an AG. The first four statements were taken from the published 2016 Eurobarometer survey on antimicrobial resistance [17]. The statements with the answers were 1. Antibiotics kill viruses (false), 2. Antibiotics are effective against cold and flu (false), 3. Unnecessary use of antibiotics makes them become ineffective (true), 4. Taking antibiotics often has side-effects such as diarrhoea (true) and 5. You can share Antibiotics with others (false). These answers were matched to the AG pledge to ascertain demographics using a direct match of email addresses. The percentage of statements answered correctly was calculated by pledge group and compared to the published results of Eurobarometer survey on Antimicrobial resistance for the UK general population and EU general population.

Behaviour change of AGs was assessed through an impact questionnaire (previously described [10]; evaluating the effect of the campaign on self-reported behaviour around AMR. The online questionnaire was completed by AG, who agreed to be followed up after pledging. Information on demographics, the motivation for becoming an AG and actions concerning pledge was collected.

The additional promotional and outcome measures were described, such as the number of registered activities during WAAW, Tweet counts, number of views of the YouTube [18] video and Playbuzz quiz on the website, number of publications, bulletins/newsletters, web articles, and international mentions.

### Data Analysis

Unless stated otherwise, Google analytics data was calculated using the number of unique visitors. Conversion rates were calculated as the proportion of visitors to the website that pledged to become an AG, and an adjusted conversion rate was calculated as the proportion of AG pledges from unique website visits. The decision to become an Antibiotic Guardian or not was calculated by the acquisition route using odds ratios and 95% confidence intervals. Significant differences between Antibiotic Guardian groups were identified using the Chi-squared test and a *p*-value of less than 0.05. Rates of antibiotics guardians per 100,000 populations were calculated with 95% confidence intervals by matching half postcodes to Clinical Commissioning Groups (CCGs) (NHS organisational bodies responsible for the planning and commissioning of health care services for their local area), health boards and local authorities in the UK and using ONS 2017 mid-year population estimates [19]. The proportion of AGs answering all four questions correctly in the knowledge survey was compared to the Eurobarometer results using odds ratios and 95% CI. Data were analysed using Microsoft^®^ Excel (2010) and STATA release 15 [20].

## 5. Conclusions

After four years the Antibiotic Guardian campaign has sustained year-round engagement within a wide range of individuals and countries, with particular increases in student and public groups. AGs (including public AGs) showed greater antimicrobial resistance knowledge when compared to the Eurobarometer UK group. Future work should focus on further promotion within the UK and Europe to sustain the engagement of the campaign and to understand and learn about differences in engagement between areas. Within England, a new media campaign for members of the public (Keep Antibiotic Working) was piloted in one region of England in 2016 and rolled out nationally in 2017. This campaign aligns with Antibiotic Guardian, and its aim will help fulfil the recommendation from the impact evaluation to reach those not previously engaged with AMR.

## Figures and Tables

**Figure 1 antibiotics-08-00021-f001:**
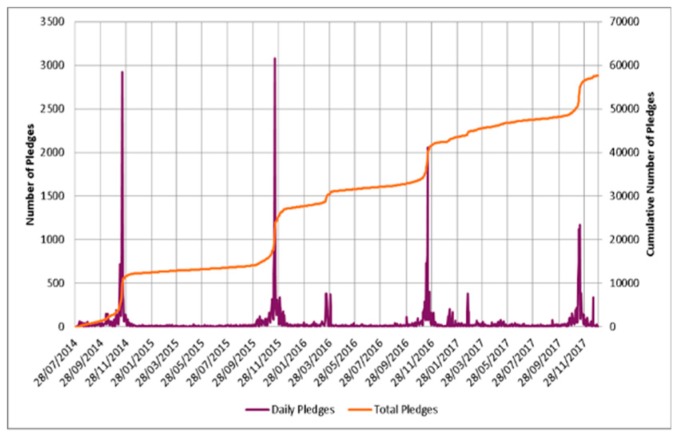
Daily number of AG pledges with the cumulative number of AG pledges from 2014 to 2017.

**Figure 2 antibiotics-08-00021-f002:**
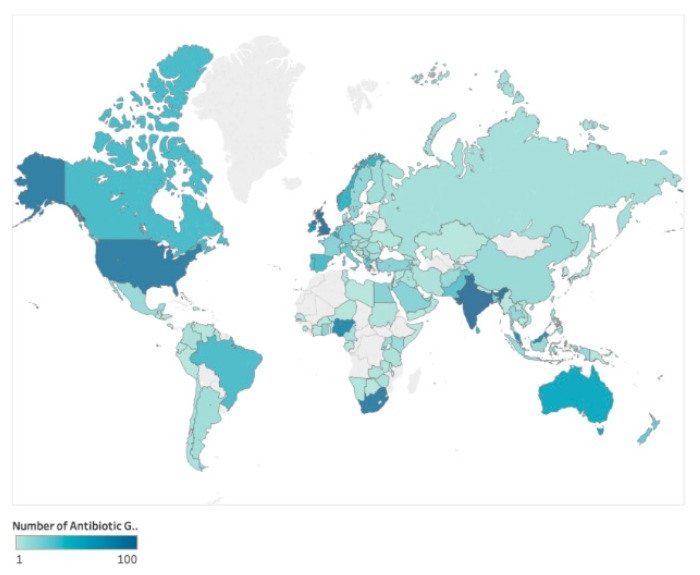
World Map of Antibiotic Guardians (2015–2017).

**Figure 3 antibiotics-08-00021-f003:**
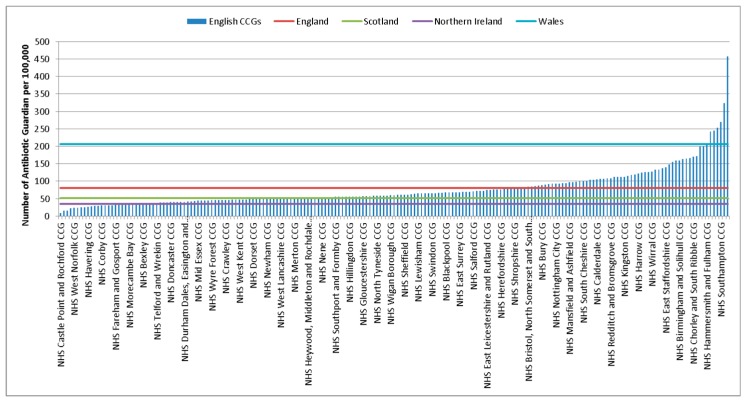
Rate of AGs per 100,000 population for English CCGs and Scottish, Welsh and Northern Ireland Health Boards (2014–2017).

**Figure 4 antibiotics-08-00021-f004:**
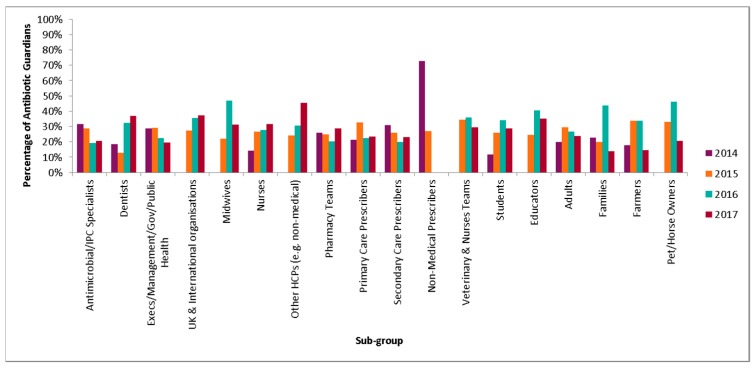
Percentage of AG pledges by sub-group from 2014–2017 (n = 56,721).

**Figure 5 antibiotics-08-00021-f005:**
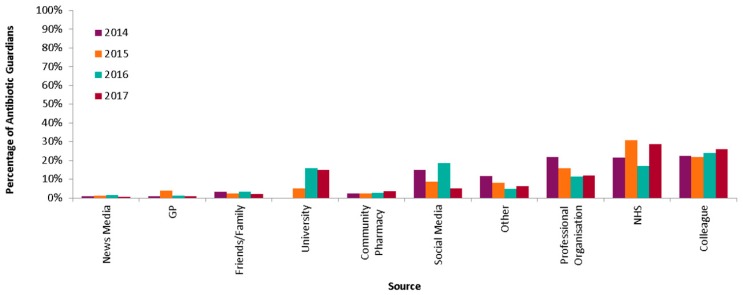
Percentage of AGs by source of hearing of the AG campaign 2014–2017.

**Figure 6 antibiotics-08-00021-f006:**
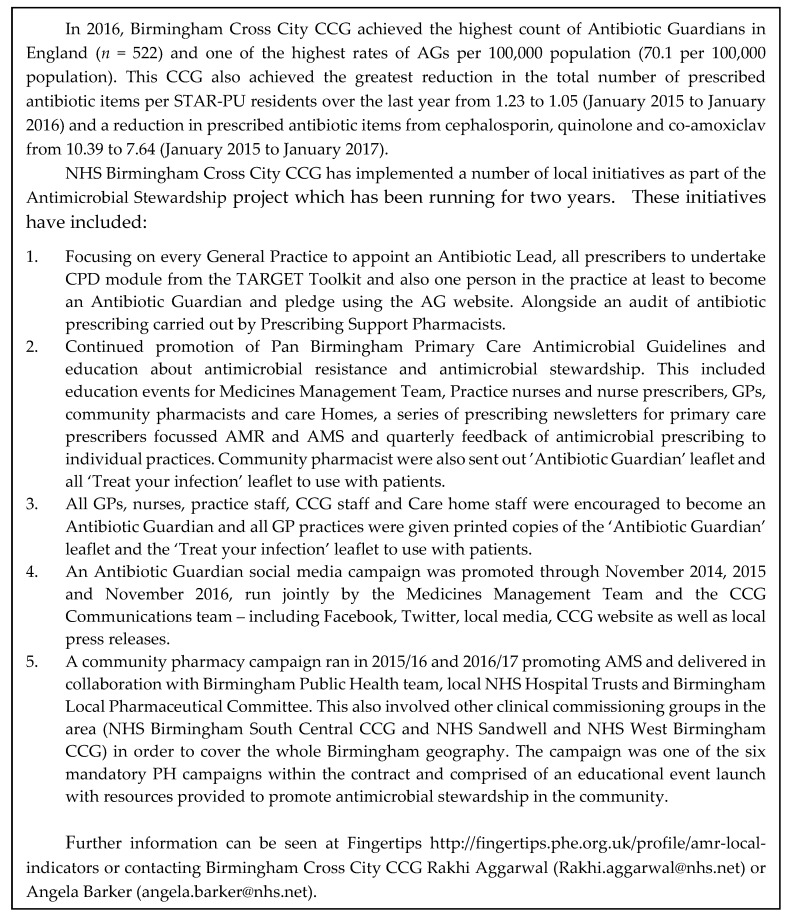
Case Study: Birmingham CrossCity CCG.

**Figure 7 antibiotics-08-00021-f007:**
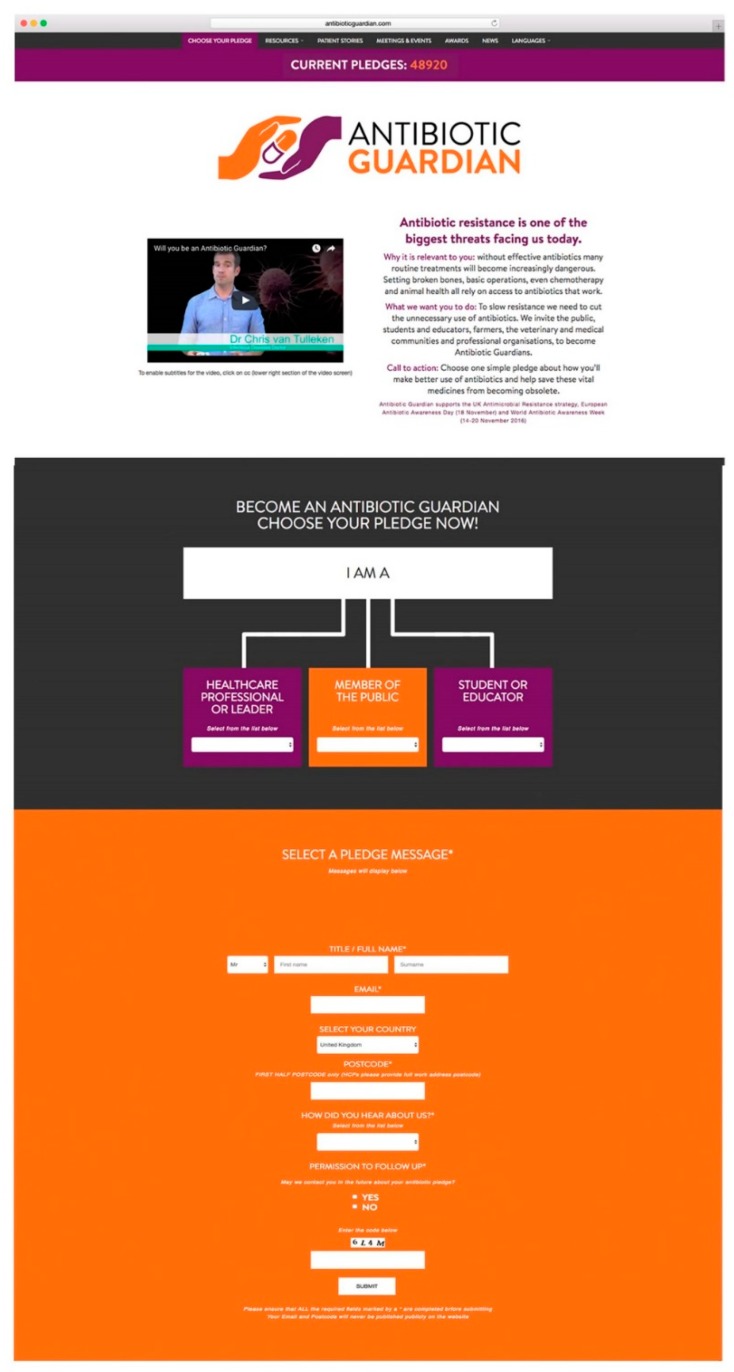
Screenshot of the Antibiotic Guardian pledge website (taken September 2017).

**Table 1 antibiotics-08-00021-t001:** Antibiotic Guardian website metrics including conversion rates from 2014–2017.

AG Metrics	2014	2015	2016	2017
Visits/Visitors	56,569	67,824	96,833	70,205
Unique Visits	46,029	54,336	78,874	52,221
AG Pledges	12,315	15,002	15,140	15,170
Conversion rate—Unadjusted	21.8%	22.1%	15.6%	21.6%
Conversion rate—Adjusted	26.8%	27.6%	19.2%	29.0%

**Table 2 antibiotics-08-00021-t002:** Percentage of AG pledges by pledge group from 2014–2017 (n = 56,921).

Antibiotic Guardian	2014 (n = 12,221)	2015 (n = 14,843)	2016 (n = 15,048)	2017 (n = 14,809)
Healthcare Professional	63.2	52.8	46	56.9
Public	28	30.2	31.4	23.8
Student and Educators *	8.8	17.1	22.5	19.4

* A broad group which included specific pledges for students and educators.

**Table 3 antibiotics-08-00021-t003:** Most common pledge by Antibiotic Guardian subgroups including number and proportion of pledges, 2014–2017.

Sub-Group	Most Common Pledge	Number of Pledges	Pledge
**Healthcare Professionals**			
Antimicrobial/IPC Specialists	I will encourage and champion members of my organisation to become Antibiotic Guardians	375	24.4%
Dentists	When I see a patient with dental pain, I will discuss methods of controlling symptoms rather than prescribing antibiotics as a first course of action	640	54.9%
Execs/Management/Gov/Commissioners/Public Health	During the cold and flu season, I will add the Antibiotic Guardian electronic signature to all my emails	552	27.6%
Midwives	When a mother is prescribed antibiotic, I will ensure that she understands why they have been prescribed and to how take them	71	42.0%
Non-Medical Prescribers	When I write an antimicrobial prescription I will make sure it’s in line with local guidelines	116	47.0%
Nurses	Every time, I see an antibiotic prescription which has continued beyond seven days without specified duration, I will highlight this to the prescriber or pharmacist	1493	23.3%
Other HCPs (e.g., non-meds)	I will encourage clients/patients and colleagues to take the online Antibiotic Guardian quiz and choose their own pledge to become Antibiotic Guardians	1170	48.5%
Pharmacy Teams	I will check that antibiotic prescriptions comply with local guidance and query those that do not	4147	39.5%
Primary Care Prescribers	I will ensure all prescribers in my practice including locums have easy access to the local antibiotic guidance	527	21.4%
Secondary Care Prescribers	If I prescribe an antibiotic then I will document indication, duration and review dates on the drug chart in line with Start Smart then Focus AMS guidance	977	158.3%
UK & International organisations	As an organisation, we will work to help develop a national action plan for our country on antimicrobial resistance in line with the global plan	36	27.3%
Veterinary Teams	If there is a need to prescribe antibiotics I will use narrow spectrum drugs wherever possible	273	24.1%
**Student and Educators**			
Students	The next time I see an antibiotic prescribed, I will ask the prescriber about the indication and duration, to understand if this is in accordance to local and national guidelines	1767	19.2%
Educators	I will encourage my students to impart their knowledge, challenge their peers to take the Antibiotic Guardian quiz online and choose a pledge to become an Antibiotic Guardian	275	41.2%
**Public**			
Adults	For infections that our bodies are good at fighting off on their own, like coughs, colds, sore throats and flu, I pledge to try treating the symptoms for five days rather than going to the GP	5211	40.5%
Families	If anyone in my family is prescribed antibiotics, I will ensure they are taken exactly as prescribed and never shared with others	617	22.6%
Farmers	I will keep my animal(s) healthy through good nutrition and husbandry, relevant vaccination and worming and by having regular veterinary health checks	18	29.0%
Pet Owners	To help reduce the need for antibiotics I will keep my animal healthy through exercise, good nutrition, relevant vaccination, and by having regular veterinary health checks	127	26.7%

**Table 4 antibiotics-08-00021-t004:** Breakdown of student Antibiotic Guardians by study area from 2014–2017 (n = 8826).

Study Area	Count	Percentage
Pharmacy	4273	48.4
Nursing	1087	12.3
Medical	1019	11.5
Veterinary	200	2.3
Dental	105	1.2
Other healthcare student	785	8.9
Non-healthcare student	1357	15.4

**Table 5 antibiotics-08-00021-t005:** Percentage of correct answers to the AMR knowledge questions by pledge group and compared to Eurobarometer.

Group	Question 1	Question 2	Question 3	Question 4	Question 5	Questions 1–4 All Correct
Healthcare professional	95	97	99	94	97	88
Students	91	90	97	90	96	71
Public	85	93	97	84	96	77
All AG	92	95	98	91	97	81
Eurobarometer UK	56	73	92	63	n/a	34
Eurobarometer EU	43	56	84	66	n/a	24

Questions: 1. Antibiotics kill viruses (false), 2. Antibiotics are effective against cold and flu (false), 3. Unnecessary use of antibiotics makes them become ineffective (true), 4. Taking antibiotics often has side-effects such as diarrhoea (true) and 5. You can share Antibiotics with others (false).

**Table 6 antibiotics-08-00021-t006:** Characteristics of respondents, number of observations (N) = 1940.

Variable	N (%)
**Pledge Group**	
Healthcare Professional	1086 (56%)
Members of Public	598 (30.8%)
Student	179 (9.2%)
Missing	77 (4%)
**Gender**	
Male	351 (18.1%)
Female	1197 (61.7%)
Prefer not to say	11 (0.6%)
Missing	381 (19.6%)
**Age**	
<44 years old	666 (34.3%)
45–74 years old	878 (45.3%)
>75 years old	7 (0.4%)
Prefer not to say	11 (0.6%)
Missing	378 (19.5%)
**Prior Knowledge on AMR?**	
Yes	1669 (86%)
No	43 (2.2%)
I can’t remember	12 (0.6%)
Missing	216 (11.1%)
**Acting in Line with Pledge After Campaign**	
Always	1124 (58%)
Most of the time	491 (25.3%)
Some of the time	56 (2.9%)
Occasionally	35 (1.8%)
Never	14 (0.7%)
I can’t remember	98 (5.1%)
Missing	122 (6.3%)

**Table 7 antibiotics-08-00021-t007:** Definition of key data analysed.

**Acquisition route ***	The route visitors use to access the website e.g., email or self-directed.
**AG Pledge**	Individual that makes a pledge on the website (antibioticguardian.com).
**Conversion Rate—Adjusted ***	The proportion of unique visitors that chose one of the pledges on the Antibiotic Guardian website to become an Antibiotic Guardian during a time period.
**Conversion Rate—Unadjusted ***	The proportion of total visitors that chose one of the pledges on the Antibiotic Guardian website to become an Antibiotic Guardian during a time period.
**Hashtag**	A type of metadata tag used on social networks such as Twitter, Facebook, YouTube. It allows users to apply dynamic and user generated tagging, making it possible for others to easily find messages with a specific theme or content. For this campaign, the hashtag is #AntibioticGuardian.
**IP address**	A unique digital signature used on the internet to identify a computer/mobile device; these can be used to distinguish geographic location and between unique and repeat visits
**Number of Publications**	number of publications including bulletins/newsletters, web articles and international mentions of Antibiotic Guardian was calculated through internet search
**Referral/Directed ***	A link that directs individuals to AntibioticGuardian.com; these can be found on another website, in an email or in a social media post and do not include self-directed traffic
**Registered Organisations**	The number of organisations who registered their planned activities for World Antibiotic Awareness Week/European Antibiotic Awareness Day with Public Health England
**Self-Directed ***	Visitor that access the website directly through the browser address field or a search engine
**Social Media Channels**	A summary term to describe online media sharing platforms such as twitter, Facebook, LinkedIn or Google+
**Tweet Count**	The number of tweets using the hashtag #AntibioticGuardian was determined from the social media analytics platform Symplur (symplur.com/healthcare-hashtags/antibioticguardian)
**Unique Visits ***	The first time an IP address is accesses the antibiotic guardian website; this is used as an indication for an Individual access with subsequent visits from the same IP address registered as repeat. This method cannot distinguish between multiple people who may share a single device as is often the case in NHS organisations in the UK.
**Visits/Visitors ***	a person access the website (antibioticguardian.com).

*—data calculated using data available through Google Analytics [16].

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
