# Peer review of "Demographic, Knowledge and Impact Analysis of 57,627 Antibiotic Guardians Who Have Pledged to Contribute to Tackling Antimicrobial Resistance"

_antibiotics, 2019, doi:10.3390/antibiotics8010021_

Round 1

Reviewer 1 Report

The manuscript is on a relevant public health topic (antimicrobial resistance- AMR) but has multiple flaws. In order to improve the quality the manuscript should be re-written in order to remove redundant data/ information and improve clarity. The majority of the tables and figures are either difficult to read or contain redundant information.

The authors split the participants in categories: HCP, public and students. The last category is not clearly defined. For exemple in table 2 in the student’s category include students and educators. This is not correct in my opinion. Furthermore, I think that the authors should clearly split the students according to the study area because it would be expected that health sciences students would have greater awareness of this subject that other students.

The authors should avoid repeat in the discussion section what have been described in the previous section (results). In order to improve discussion more references to other studies should be included.

In the methods section the authors should correct the duration of the study. The references should also be checked with emphasis on reference 15/16 that are meaningless.

Author Response

The manuscript is on a relevant public health topic (antimicrobial resistance- AMR) but has multiple flaws. In order to improve the quality the manuscript should be re-written in order to remove redundant data/ information and improve clarity. The majority of the tables and figures are either difficult to read or contain redundant information.

Thank you very much for your comments. We have rewritten and reordered several sections of the manuscript to improve clarity. The key areas are highlighted.  Specifically:

Result  section has been reorganised and with an added sub-section ‘Engagement’ to      improve the clarity and flow for the readers.

Redundant information deleted from the result and discussion section

The authors split the participants in categories: HCP, public and students. The last category is not clearly defined. For exemple in table 2 in the student’s category include students and educators. This is not correct in my opinion. Furthermore, I think that the authors should clearly split the students according to the study area because it would be expected that health sciences students would have greater awareness of this subject that other students.

We have clarified the student and educator issue within the table and updated the text (line 156-158). Table 2 shows the 3 main categories on the website from which more specific category is chosen. Students and educators are grouped together and then from this, specific educator and student pledges could be chosen. We have added a screenshot of the website in the manuscript to increase clarity (figure S1). We have also included they breakdown of students by study area (table 2a). We do not have data on student split for knowledge questionnaire. The content of tables 1 to 4 have also been updated, focusing on key information

The authors should avoid repeat in the discussion section what have been described in the previous section (results). In order to improve discussion more references to other studies should be included.

Thank you for this comment. We have as highlighted earlier rewritten and reordered several parts of the manuscript. Specifically, we have removed repetition of results in discussion section –Lines 231-239, 249, 265-266 in original manuscript submitted

In the methods section the authors should correct the duration of the study. The references should also be checked with emphasis on reference 15/16 that are meaningless.

Thank you for this and our apologies. Ref 15/16 were part of the template provided which we had not removed. We have also updated the methods – line 320

Reviewer 2 Report

The introduction of the manuscript is very limited in scope, it would be of importance to readership if the authors included additional information. Especially the first paragraph could be expanded to quantify the extent of the problem in the target region as well as globally. Since North American readers are an important part of Antibiotics readership and contributors to the current data as participants, relevant information from CDC could also be briefly mentioned as well such as:

https://www.cdc.gov/drugresistance/biggest_threats.html

 The important information about the completion rate and properties of the data instrument, methods and software used for analysis, and data management would need to be provided. These pieces of information are critical to make a research article repeatable in the future. The current study is rather promoting an awareness campaign and does not provide fundamental information required in a manuscript. Also, authors indicate their study is a cross-sectional study, depending on when the data was analyzed this study is a prospective or retrospective cohort study.

The manuscript discusses educational YouTube video digital footprint but no information is provided amount the content and reason for developing such material. Similarly, information about twitter campaign on line 191 is not provided. Methods and material section must be completed in a fashion that the studies become a repeatable endeavor for future scientists working on similar campaigns.

Some of the graphical representation currently do not have the publication quality. Table 4 as an example could be constructed in Microsoft word with proper alignment, also tables could be harmonized in style e.g. table 4 format could resemble those of previous tables.

Some of the references appear to be only listed in the references section but not cited in the text, for example, the reviewer was unable to locate were the references 11 to 14 are cited in the text.

Author Response

The introduction of the manuscript is very limited in scope, it would be of importance to readership if the authors included additional information. Especially the first paragraph could be expanded to quantify the extent of the problem in the target region as well as globally. Since North American readers are an important part of Antibiotics readership and contributors to the current data as participants, relevant information from CDC could also be briefly mentioned as well such as:

https://www.cdc.gov/drugresistance/biggest_threats.html

Thank you very much for your comment. We have expanded the introduction to highlight the global challenge and included additional references.

 The important information about the completion rate and properties of the data instrument, methods and software used for analysis, and data management would need to be provided. These pieces of information are critical to make a research article repeatable in the future. The current study is rather promoting an awareness campaign and does not provide fundamental information required in a manuscript. Also, authors indicate their study is a cross-sectional study, depending on when the data was analyzed this study is a prospective or retrospective cohort study.

Thank you for your comments. We have added definition of key data analysed (table 6) to the methods section to improve clarity and support reproducibility. We have also updated the methods to reflect that this is a cohort study.

The manuscript discusses educational YouTube video digital footprint but no information is provided amount the content and reason for developing such material. Similarly, information about twitter campaign on line 191 is not provided. Methods and material section must be completed in a fashion that the studies become a repeatable endeavor for future scientists working on similar campaigns.

Thank you very much for this valid point. We have provided additional information in lines 52 to 62 providing further information about the YouTube digital footprint and use of Twitter.

Some of the graphical representation currently do not have the publication quality. Table 4 as an example could be constructed in Microsoft word with proper alignment, also tables could be harmonized in style e.g. table 4 format could resemble those of previous tables.

Thank you, we have updated all tables including table 4.

Some of the references appear to be only listed in the references section but not cited in the text, for example, the reviewer was unable to locate were the references 11 to 14 are cited in the text.

We have updated all references.

Reviewer 3 Report

The paper is really helpful to understand and analyze the awareness in human to the antibitotic resistance. I wish to see the proportion of the people taking the pledge increases year by year. Great work!!

Author Response

The paper is really helpful to understand and analyze the awareness in human to the antibitotic resistance. I wish to see the proportion of the people taking the pledge increases year by year. Great work!!

Thank you very much for your comment.

Round 2

Reviewer 1 Report

In this new version, the manuscript is improved and questions previously raised were answered. Nevertheless, in my opinion the manuscript should be checked for formatting issues. The S1 table is not quoted in the text and in table 4, questions were numbered but neither the question nor the answers are available. This last aspect is important for the reader.

Author Response

Thank you very much for the second review and thank you for taking the time to share additional feedback which we have now implemented:

The S1 table is now quoted in the text. (line 124)

We have also added the questions/key to table 4. (line 207/208)

Reviewer 2 Report

Authors have improved the repeatability of their study by providing additional information in the Methods and Materials section and have incorporated key references. 

Author Response

Thank you very much for your comments.